# Scaling Laws for Mixed quantization in Large Language Models

## Abstract

Post-training quantization of Large Language Models (LLMs) has proven effective in reducing the computational requirements for running inference on these models. In this study, we focus on a straightforward question: When aiming for a specific accuracy or perplexity target for low-precision quantization, how many high-precision numbers or calculations are required to preserve as we scale LLMs to larger sizes? We first introduce a critical metric named the quantization ratio, which compares the number of parameters quantized to low-precision arithmetic against the total parameter count. Through extensive and carefully controlled experiments across different model families, arithmetic types, and quantization granularities (e.g. layer-wise, matmul-wise), we identify two central phenomenons. 1) The larger the models, the better they can preserve performance with an increased quantization ratio, as measured by perplexity in pre-training tasks or accuracy in downstream tasks. 2) The finer the granularity of mixed-precision quantization (e.g., matmul-wise), the more the model can increase the quantization ratio. We believe these observed phenomena offer valuable insights for future AI hardware design and the development of advanced Efficient AI algorithms.

## 1 Introduction

Large Language Models (LLMs) have demonstrated remarkable performance across a range of natural language processing (NLP) tasks (Brown et al., 2020), and state-of-the-art models have ranged from 1.6B parameters (Radford et al. (2019)) to 1T parameters (Fedus et al. (2022)) in recent years. Recent work has driven the development of even larger models given findings that LLMs exhibit emergent capabilities at increased parameter counts (Wei et al., 2022a). As such, researchers have endeavoured to understand the scaling laws of LLMs by characterising how the required number of training tokens scales with parameter count to train compute-optimal models under a fixed compute budget (Kaplan et al. (2020), Hoffmann et al. (2022)). These works provide insight on how to best allocate resources in training increasingly large LLMs.

Despite recent scaling trends, the substantial size of LLMs and their accompanying computational demands require significant energy and hardware resources. For instance, inference deployment of the 405B parameter LLaMA-3.1 model (Dubey et al., 2024) requires 8 NVIDIA H100 GPUs to store its 810GB of model weights, and consumes over 4500 Watts of power (based on the average power consumption of 600W per H100 GPU). As such, quantization is emerging as a promising solution to increase the accessibility of LLMs by reducing their memory requirement and inference cost. Prior work has shown that weights and activations in pretrained transformer blocks often yields magnitude outliers, which has been addressed by casting outliers to high precision, while quantizing the rest of the network to low precision (Dettmers et al., 2022). Such mixed-precision partitioning has been shown to preserve model performance with significant savings in memory footprint and model inference serving throughput.

With the increased usage of quantization to address the challenges of LLM deployment, and motivated by the importance of understanding systematic scaling laws in guiding further research in mixed-precision quantization, we seek to answer an under-explored question: in an optimal mixed-precision mapping, how does the required ratio of low precision components change as model size increases? Alternatively, *what are the scaling laws for mixed quantization in LLMs*?

We define the mixed-quantization ratio $Q_r$ as the ratio of parameters using low-precision arithmetic to the total number of parameters (i.e. $\frac{\text{Low Precision Parameters}}{\text{Total Parameters}}$), and consider the scenario where no finetuning takes place after quantization. To illustrate our main results, we show how performance scales with both model size and mixed-quantization ratios for Qwen models in Figure 1. The figure demonstrates our key observation that as model size increases, higher quantization ratios yield a lower performance penalty. In fact, through extensive and carefully controlled experiments, we show that *the number of low-precision components*

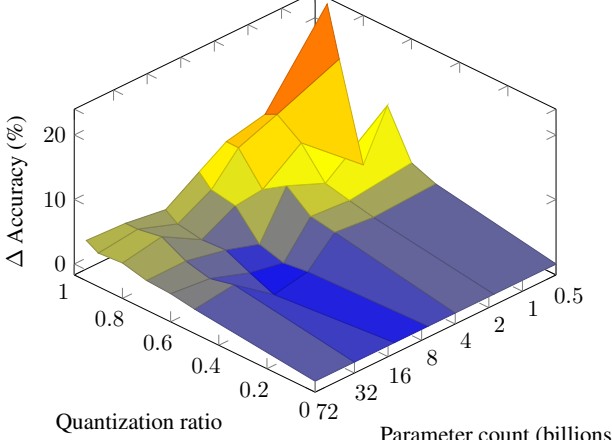

Figure 1: Change in accuracy on the MMLU dataset for models in the Qwen family quantized to MXINT4 at various Quantization Ratios, defined as $\frac{\text{Low Precision Parameters}}{\text{Total Parameters}}$.

*scales exponentially relative to the growth in model size* under a fixed performance budget.

Additionally, we examine the practical aspects of deploying mixed-precision LLMs, namely the granularities at which quantization can be applied (i.e. per transformer layer, matrix multiply operation, etc), and how this affects performance degradation as parameter count increases. We find that mixed-precision LLMs benefit greatly from quantization at finer granularities, by effectively leveraging the unstructured distribution of outliers in weights and activations.

Our main contributions are as follows.

- We conduct a series of carefully designed, controlled experiments across various model families, arithmetic types, and quantization granularities to examine the scaling behaviour of LLMs in the context of mixed-precision quantization.

- We summarize the results and formulate two scaling phenomena, named *LLM-MPQ* scaling laws, for mixed quantization in LLMs.

- We discuss the potential benefits and implications of the proposed scaling laws for future AI inference systems and hardware designs, arguing that the advancement of low-precision arithmetic hardware could facilitate the scaling of future LLM inference.

## 2 BACKGROUND

### 2.1 QUANTIZATION OUTLIERS AND MIXED QUANTIZATION

**Weight and Activation Outliers in LLMs** A weight or activation value is considered an outlier when there is a significant deviation from its mean distribution. These values make traditional uniform quantization less effective due to the large incurred dynamic range. In fact, activation outliers have been observed more frequently in large transformer-based models (Wei et al., 2022b; Zhang et al., 2023a) as deeply cascaded layers accumulate quantization errors. To address this problem, two strategies are widely adopted; in weight-only quantization, weights are casted to low precision while activations are left in a higher-precision format such as FP16 or BF16 (Frantar et al., 2022; Chee et al., 2024; Lin et al., 2024). Meanwhile, weight-activation quantization relies on methods such magnitude transfer from activations to weights using invertible scale matrices (Xiao et al., 2023) to alleviate the effect of activation outliers before quantizing both weights and activations (Wei et al., 2023; Xiao et al., 2023; Shao et al., 2023).

Recent works have proposed novel numerical formats with shared scaling/exponent components, which better accommodate the dynamic range of outliers (Zhang et al., 2023a; Rouhani et al., 2023a; Zou et al., 2024). For example, MXINT (Darvish Rouhani et al., 2020) is new standard for hardware-efficient numerical formats sharing an exponent across a block of mantissas (Rouhani et al., 2023b). The hardware efficiency of these methods often outperforms standard low-precision floating-point computation, although custom hardware support is required to be used in practice.

**Mixed-precision quantization** Mixed precision approaches involve partitioning a model's parameters into both high-precision and low-precision components, which have been shown to better preserve model performance relative to uniform quantization. This is primarily seen in models that exhibit different sensitivities to quantization at various layers. Some mixed-precision LLM quantization work adopts the concept of weight salience to guide the search for fine-grained bit allocation. The first order (Li et al., 2023a) or second order weight gradient (Huang et al., 2024) have been used to form such salience metrics, such that salient layers are left in higher precision while the rest are casted to low precision. There are also works performing the search in an end-to-end style with the quantized model performance as the objective, such as the accuracy on a downstream task (Zhang et al., 2023a). In both cases, mixed-precision can be seen as a promising approach to provide lossless reduction in LLM memory requirements, reducing average bitwidths below levels achievable through uniform quantization (as in Badri & Shaji (2023); Lin et al. (2024); Chee et al. (2024)).

**Mixed-precision inference** Mixed-precision inference methods targeting GPUs usually adopt regular mixed-precision strategies and computation patterns; The authors of GPTQ3.int8() (Dettmers et al., 2022) decompose the matrix multiplication in every linear layer into two submatrix-multiplications based on the activation magnitudes, achieving 2-3× inference speedup by casting the low-magnitude submatrix to low precision. SpQR (Dettmers et al., 2023) represents a weight matrix with grouped 3-bit integers and less than 1% sensitive weight elements with FP16 values, achieving 2× speed up compared to a quantized and sparse PyTorch baseline. These approaches enable reducing model size, but additional careful treatment is needed to improve inference throughput. For example, in Li et al. (2023a), mixed precision LLM quantization at 2-bit and 3-bit showed no speedup compared to 4-bit, due to less efficient utilization of memory bandwidth. On the other hand, Any-Precision LLM (Park et al., 2024) achieve throughput scaling at various precisions by providing CUDA kernels with a novel weight packing approach following a bitplane layout, achieving 1.3-1.8× speedup on mobile and edge devices. Additionaly, works such as FlightLLM achieve high throughput by leveraging custom hardware designs (Zeng et al., 2024).

**Mixed-precision training** Mixed-precision quantization has also been adopted in training to reduce the large memory footprint of gradient descent, which requires storage of optimizer states and gradients in addition to forward activations. It's been shown the training process can tolerate aggressive quantization and correct quantization noise in some components. Micikevicius et al. (2017) is the pioneering work proposing storage of all weights, activations and gradients in FP16, while updating a copy of weights in FP32. This work also proposed scaling up the forward pass loss and unscaling the gradient before weight update to avoid underutilization of the FP16 representable range, leading to half the memory requirement and speed-ups of 2-6× relative to FP32 training.

Recently, more aggressive quantization has been studied for mixed-precision training. Mellempudi et al. (2019) trains models with E5M2 FP8, maintaining a master copy of the weights in FP16, and dynamically adjusting the scaling factor every few iterations. Hybrid-FP8 (Sun et al., 2019) improves FP8 training by using E4M3 for forward propagation and E5M2 for backward propagation, leading to matching performance to models trained with FP32. Popular implementation of FP8 mixed-precision training like TransformerEngine[1] has achieved a training acceleration around 3-4× compared to FP16 mixed-precision training.

## 2.2 Scaling Laws of Large Language Model Training

Enhanced language modelling performance at larger model sizes has led to interests in characterising scaling laws of LLM performance with respect to parameter count, training compute budget and number of training tokens. Kaplan et al. (2020) showed, though empirical analysis, that Transformer performance follows a power law trend with each of these factors. The authors proposed that for any 9× increase in parameter count, dataset size should be increased by a factor of approximately 5× to avoid a performance penalty, suggesting that higher performance gains are observed from scaling parameter count than dataset size under a fixed compute budget. Contrastingly, Hoffmann et al. (2022) argue that existing open weight LLMs are undertrained relative to their size, and parameter count should be increased in line with the number of training tokens. The findings from Pearce & Song (2024) later explained the discrepancy between Kaplan and Hoffman, reaffirming the validity

---

[1]TransformerEngine: https://github.com/NVIDIA/TransformerEngine

of the Chincilla scaling laws and highlighting the need for high quality datasets for Language Model training.

Despite the reconciliation of Kaplan's scaling laws, we see the continued trend of scaling LLM sizes, partly validated by the findings from Wei et al. (2022a) regarding the unpredictable emergence of abilities in larger LLMs. The authors characterise how LLM performance in few-shot tasks including arithmetic, question answering and multi-task language understanding emerges beyond certain size thresholds, despite not being observable in small models - for example, performance on arithmetic tasks from BIG-Bench is approximately random for GPT-3 models up to 13B parameters and LaMDA models up to 68B parameters, sharply rising thereafter. The authors note there are few compelling explanations for these emergence phenomena, although the required model depth for reasoning tasks may be correlated with the number of reasoning steps. At any rate, these findings raise the question of what emergent phenomena may observed for even larger models, and highlight the importance of understanding scaling laws across a wide range of model sizes, without extrapolating observations from small models.

## 3 SCALING LAWS FOR MIXED QUANTIZATION

Consider a model $F(W_l, W_h)$, parameterized by low and high-precision components, $W_l$ and $W_h$. We define a model's quantization ratio $Q_r$ as the ratio of low-precision parameters to the total number of parameters, i.e. $\frac{\|W_l\|_0}{\|W_h\|_0 + \|W_l\|_0}$, where $\|\cdot\|_0$ computes the $l_0$ norm. The optimal allocation of low-precision parameters for a model under a fixed quantization ratio, described by $W_l^{opt}$ and $W_h^{opt}$, can be found through the following optimization problem, where $L(\cdot)$ is the task loss.

$$W_l^{opt}, W_h^{opt} = \underset{W_l, W_h; s.t. \frac{\|W_l\|_0}{\|W_h\|_0 + \|W_l\|_0} = Q_r}{\arg\min} L(F(W_l, W_h)) \qquad (1)$$

Equation 1 outlines the optimization problem used to evaluate the hypothesized scaling laws. In this work, we find an approximate solution to the problem using a random search algorithm to allocate a numerical precision to each component of the network (i.e. layer or matrix multiply operation, according to the granularity). Furthermore, no weight training is performed after quantization, such as to observe the immediate performance degradation. We describe the observed scaling laws in this section, and present empirical evidence to support them in Section 4.

---
**_LLM-MPQ_ Scaling Law 1: Scaling with Model Sizes**

Given a fixed loss budget $L_{max}$, the maximum achievable mixed precision quantization ratio $Q_r = \frac{\|W_l\|_0}{\|W_h\|_0 + \|W_l\|_0}$ increases as the model size ($\|W_h^{opt}\|_0 + \|W_l^{opt}\|_0$) increases.

---

The first scaling law posits our central hypothesis: as model size grows, so does the required quantization ratio, under a fixed task loss target. This aligns with findings from related research, such as AWQ (Lin et al., 2024), Quip (Chee et al., 2024) and LQER (Zhang et al., 2024), which empirically demonstrated that larger models can accommodate more aggressive quantization levels. An alternative view, also reflected in related work, is that for a fixed quantization ratio, task loss decreases when the model size becomes larger.

---
**_LLM-MPQ_ Scaling Law 2: Scaling with Quantization Granularities**

Given a fixed loss budget $L_{max}$, the maximum achievable mixed precision quantization ratio $Q_r = \frac{\|W_l\|_0}{\|W_h\|_0 + \|W_l\|_0}$ increases if a finer granularity is applied to $W_l^{opt}$ and $W_h^{opt}$.

---

The second scaling law focuses on the granularity of quantization, which can refer to the size of the group in which quantization is applied, *e.g.*, per-vector, per-tensor or per layer. This hypothesis is reflected by observations from previous studies like Dettmers et al. (2022), which noted that specific parameter groups required high-precision components to avoid performance degradation. At lower quantization granularities, a more significant portion of the high-precision quantization budget is

allocated to operations that could be casted to low precision without a performance penalty. This effect is particularly pronounced when the distribution of outliers is highly irregular.

Many studies have empirically demonstrated that larger models are more amenable to quantization (Dettmers et al., 2022; Xiao et al., 2023), and in this work, we offer a systematic perspective on this finding by formulating the aforementioned Scaling Laws. Crucially, we illustrate that model size (Law 1) **exhibit exponential scaling relative to the "ease of quantization"** while quantization granularity (Law 2) **exhibit power function scaling relative to the "ease of quantization"** in a mixed-precision setting. These observation suggest that the hidden law demonstrated under mixed quantization settings are non-trivial. Here, we refer to the "ease of quantization" as the proportion of high-precision components necessary to maintain model performance.

## 4 EXPERIMENTS

### 4.1 EXPERIMENT SETUP

**Models and benchmarks**   We evaluate a range of model families, including LLama-2 (Touvron et al., 2023), Gemma-2 (Team et al., 2024) and QWen-1.5 (Bai et al., 2023), at sizes ranging from 0.5B to 70B. We consider both pre-trained and instruction-tuned models. The primary results are reported for QWen-1.5 as a wide range of pretrained checkpoints is available, enabling detailed analysis of the proposed scaling laws. Further results are included in Appendix A.

We evaluate LLMs on WikiText2 (Merity et al., 2016), SlimPajama (Soboleva et al., 2023), Alpaca (Taori et al., 2023), and MMLU (Hendrycks et al., 2021b;a). For WikiText2, SlimPajama, and Alpaca, we sub-sample the dataset used during search. For MMLU, we use `lm-evaluation-harness` (Gao et al., 2023) to report the average accuracy over all subsets.

**Quantization methods**   We use MXINT-4 as the low-precision format and BF16 as the high precision format. We also consider FP4 for low precision, which has garnered increased attention in recent works (Liu et al., 2023b; Xia et al., 2024b; Zhang et al., 2023b; Xia et al., 2024a). For FP4, we use 2-bit exponent and 1-bit mantissa, which offers the highest performance without any post-quantization fine-tuning, as per Dotzel et al. (2024). We quantize both weights and activations for MXINT-4, however activations are kept in BF16 for the FP4 experiments as activation quantization was found to cause a detrimental effect in model performance at this precision (Liu et al., 2023a).

**Mixed-precision strategy**   For our primary experiments, we perform quantization at two granularities: layer-wise and matmul-wise. In the former, the quantization ratio is determined by the number of transformer layers casted to low precision. In the latter, we consider the precision for each individual matrix multiplication; for example, QKV projections, multiplication of the attention scores and MLP layers can each be quantized separately, even within the same layer. We find a solution for Equation (1) by running random search with a trial number of 50 at 1024 subsamples per iteration. We justify these choices in Appendix D and Appendix E. The inner loop of the search conducts post-training quantization (PTQ), and the entire search process involves no training.

**Platform and GPU hours**   We perform experiments on a 20-node cluster with eight A6000 48GB GPUs, a 256-core AMD EPYC processor and 1024GB RAM in each node. Experiments of models larger than 30B are performed on a cluster of DGX A100 eight-GPU pods. The effective run time of the experiments in total is approximately 15k A6000 GPU hours and 5k A100 GPU hours. We also spend around 1k GPU hours tuning search hyper-parameters, such as the number of trials.

### 4.2 SCALING LAW 1: SCALING WITH MODEL SIZES

Firstly, we evaluate perplexity on the SlimPajama dataset at various quantization ratios and granularities to illustrate the overall loss landscape. This was chosen as the principal search task as perplexity on Alpaca showed less variance with varying quantization ratios, especially for larger models. [2]

For clarity, in Figure 2 and Figure 3 we plot the highest quantization ratio achievable at each granularity (i.e. 0.95 for layer wise and 0.99 for matmul wise) as well as a more modest quantization ratio

---

[2]A more detailed comparison between Alpaca and SlimPajama tasks is shown in Appendix C.

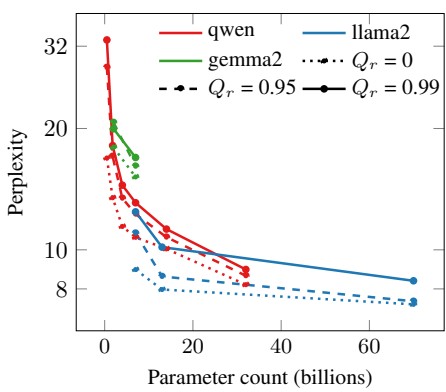 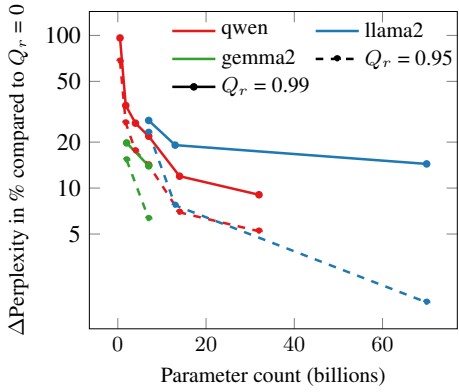

(a) Absolute perplexity on SlimPajama at varying quantization ratios.

(b) Percentage change in perplexity at varying quantization ratios, relative to non-quantized baseline.

Figure 2: Results supporting *LLM-MPQ* Scaling Law 1 at matmul granularity. We show how perplexity on SlimPajama scales with increasing model sizes under various quantization ratios ($Q_r$ values). Larger models can tolerate higher quantization ratios.

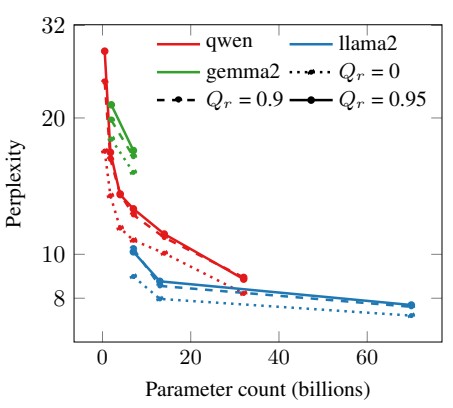 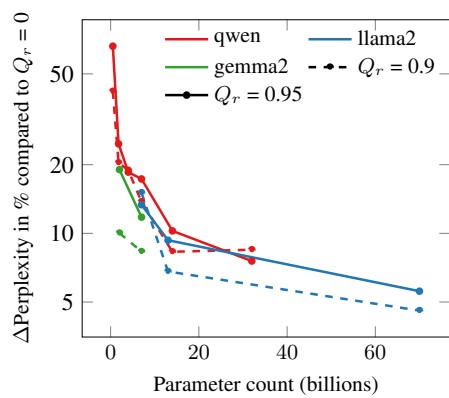

(a) Absolute perplexity on SlimPajama at varying quantization ratios.

(b) Percentage change in perplexity at varying quantization ratios, relative to non-quantized baseline.

Figure 3: Results supporting *LLM-MPQ* Scaling Law 1 at layer granularity. We show how perplexity on SlimPajama scales with increasing model sizes under various quantization ratios ($Q_r$ values). Larger models can tolerate higher quantization ratios.

(0.9 for layer wise and 0.95 for matmul wise), comparing to the non-quantized model ($Q_r = 0$) in each case. We also present zero-shot accuracy results on MMLU in Figure 4 and Figure 5, which follow a similar pattern, although more variability can be seen due to the potential bias introduced in the downstream task, although the trend can still be observed.

Both experiments demonstrate that under a fixed quantization ratio, performance metrics improve for both mixed precision strategies as the model gets larger, supporting our first scaling law. To eliminate the natural scaling law effect of increased language modelling capability at larger model sizes, we evaluate the difference in perplexity compared with the unquantized version of each model, i.e. $\Delta\text{perplexity} = (\text{ppl}_q - \text{ppl}_{\text{ori}})/\text{ppl}_{\text{ori}}$. As shown in Figure 2 and Figure 3, from both the absolute perplexity perspective and $\Delta$Perplexity perspective, the observed trend aligns with Scaling Law 1. More comprehensive results for SlimPajma perplexity are presented in Appendix A.

Although the experimental results support the first scaling law, it naturally leads to the question of the rate at which this scaling occurs. To further demonstrate our observation regarding Scaling Law 1, we can consider maximum achievable quantization ratio $C_{max}$ under a maximum performance loss budget $L_{max}$. Using perplexity change at matmul granularity for the Qwen-1.5 model, we

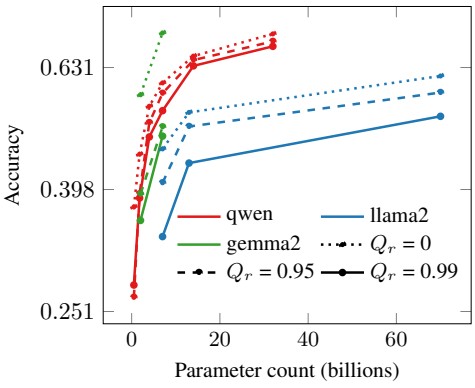
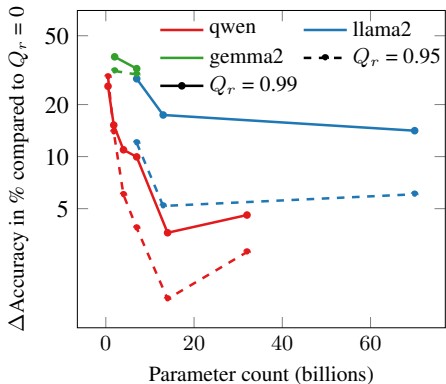

(a) Accuracy on MMLU at varying quantization ratios.

(b) Percentage change in accuracy at varying quantization ratios, relative to non-quantized baseline.

Figure 4: Results supporting *LLM-MPQ* Scaling Law 1 at matmul granularity. We show how accuracy on MMLU scale with increasing model sizes under various quantization ratios ($Q_r$ values). Bigger models can tolerate higher quantization ratios.

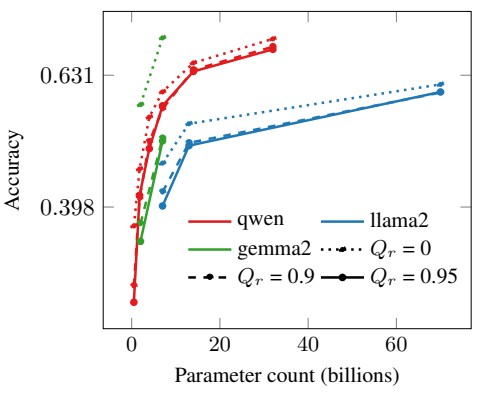
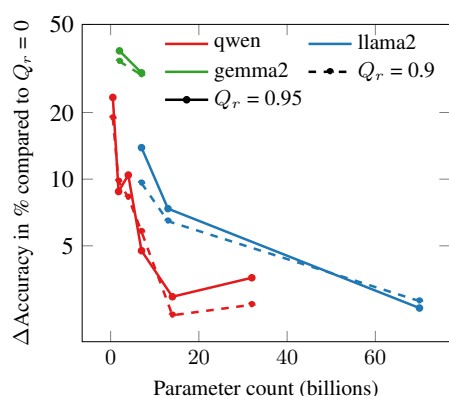

(a) Accuracy on MMLU at varying quantization ratios.

(b) Percentage change in accuracy at varying quantization ratios, relative to non-quantized baseline.

Figure 5: Results supporting *LLM-MPQ* Scaling Law 1 at layer granularity. We show how accuracy on MMLU scale with increasing model sizes under various quantization ratios ($Q_r$ values). Bigger models can tolerate higher quantization ratios.

use an exponent model to fit model size about the maximum quantization ratio under various loss budgets $L_{max}$, i.e. $y = e^{kC_{max}+c}$. As shown in Figure 6, we find that for fixed perplexity changes of 5% 10% and 20%, the obtained parameters were ($k = -11.68, c = 4.83$), ($k = -11.27, c = 3.35$) and ($k = -12.84, c = 1.86$), respectively.

In Figure 6, we use $(1 - Q_r)$ on the x-axis to represent the percentage of high-precision components required. The findings indicate that larger models require **an exponentially reduced number of high-precision components** for achieving a fixed model performance target (in perplexity or accuracy). It is also worth noting that the $k$ values of the three fitted lines in Figure 6 are in close proximity, suggesting that scaling under various model performance constraints is consistent. This underscores the significance of Scaling Law 1, as it may suggest the future requirements for low-arithmetic computation could increase exponentially with model size growth. We thoroughly examine the potential implications of these laws in Section 5.

### 4.2.1 EXTENDING TO OTHER ARITHMETIC FORMATS

We also show that the proposed *LLM-MPQ* Quantization Scaling Laws can be extended to different arithmetic formats by demonstrating an example of FP4-E2M1 (floating-point 4-bit with 2-bit exponent and 1-bit mantissa), as discussed in Section 2.1. This format is more compact than MX-

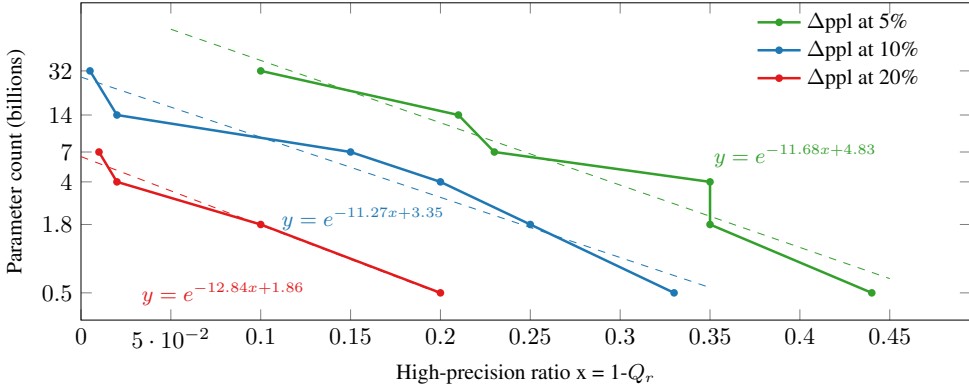

Figure 6: Fitted exponential models of model size around quantization ratio under various loss budgets. y-axis is in log-scale.

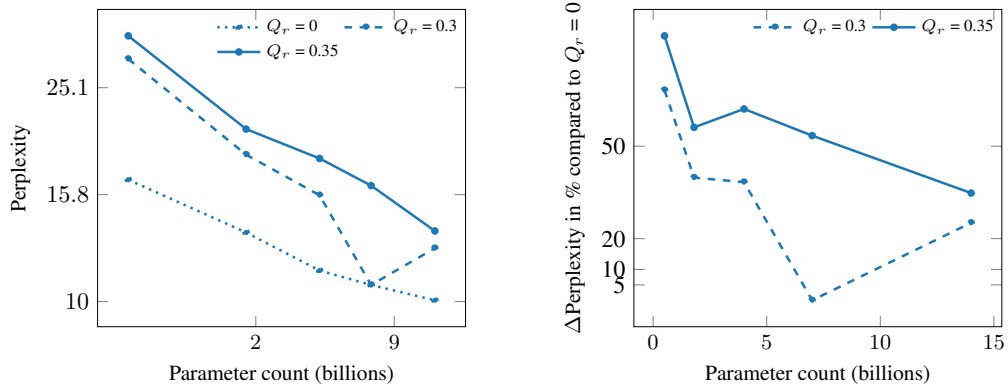

(a) Absolute perplexity on SlimPajama at varying quantization ratios.

(b) Percentage change in perplexity at varying quantization ratios, relative to non-quantized baseline.

Figure 7: Results supporting *LLM-MPQ* Scaling Law 1 for FP4-E2M1 precision.

INT4 due to the shared exponent in MX formats, but offers a smaller dynamic range and resolution. As proposed by Dotzel et al. (2024), we consider weight-only quantization (with activations kept at 16-bit) for the precision allocation search. The results are presented in Figure 7. Although the quantization ratio is generally lower than that in Section 4.2 due to the limited dynamic range and resolution of FP4, the observed trend follows *LLM-MPQ* Scaling Law 1.

### 4.3 SCALING LAW 2: SCALING WITH QUANTIZATION GRANULARITIES

As stated in Section 2.1, recent LLM quantization methods adopt fine-grained quantization, meaning tensors are split into small blocks, quantized and then scaled individually. In this subsection, we empirically verify our Scaling Law 2 by performing mixed-precision quantization search at block sizes of 16, 32, 64, 128, 256, and 512. Additionally, we perform per-vector (per-row, per-column) scaled quantization (Dai et al., 2021) and per-tensor scaled quantization. In per-vector scaled quantization, each row of activations (corresponding to a column of weights) shares the same scaling factor. In per-tensor scaled quantization, all elements in a tensor share the same scaling factor. To fit per-vector and per-tensor quantization to the same plot, we consider the averaged number of elements in each vector or tensor as the block size.

Smaller block sizes enable lower quantization error and better model performance, since a block's scaling factor depends on the maximum element magnitude within the block. When a large scaling factor is assigned to accommodate the activation outliers (as discussed in Section 2.1), the round-off error of remaining elements in the block can cause performance degradation. However, decreasing

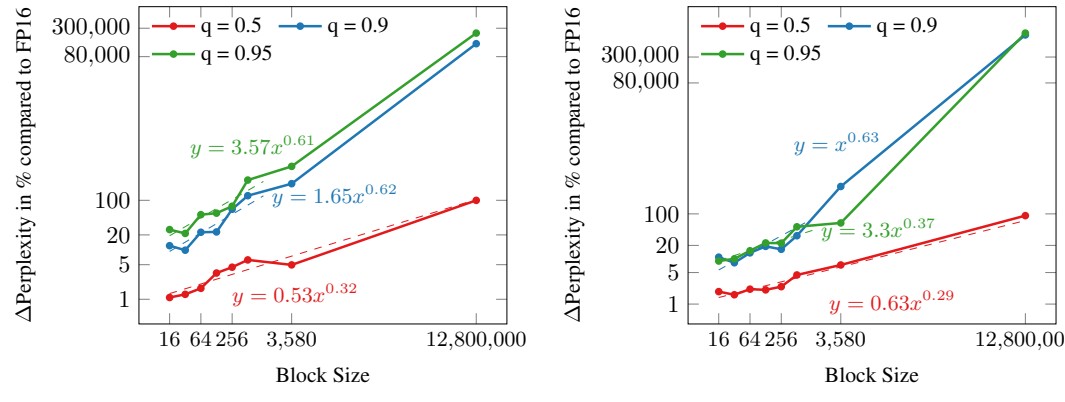

(a) Performance degradation of Qwen-1.5-7B.          (b) Performance degradation of Qwen-1.5-14B.

Figure 8: Performance degradation of Qwen-1.5 models on SlimPajama for block-wise quantization at various quantization ratios. We only fit the data points with perplexity changes within 100%. Both a-xis and y-axis are in log-scale.

the block sizes leads to higher average bitwidth, highlighting the trade-off between memory footprint and model performance in quantization granularity.

We aim to find a quantitative scaling law with quantization granularity for LLMs, that is, to inspect how the model performance changes with granularity. Figure 8 illustrates the perplexity change for the QWen-1.5-7B and 14B model relative to FP16 precision across various block sizes under various quantization budgets. In the figure, we observe the following trends:

- The perplexity change ($\Delta\text{ppl} = (\text{ppl}_q - \text{ppl}_{\text{ori}})/\text{ppl}_{\text{ori}}$) increases with block size, i.e. higher granularity contributes to lower performance degradation.
- Given a target $\Delta\text{ppl}$, smaller block sizes enable a larger quantization ratio.

To further illustrate Scaling Law 2, we fit a power function model of $\Delta\text{ppl}$ with respect to the unified the block size under various granularities (i.e. $y = Ax^k$) to show the impact of quantization granularity on model performance under various quantization ratios. Note that for quantization ratios beyond $0.95$, we observe an explosion in $\Delta ppl$, hence these values are excluded from the fitted model. We find the following model parameters for ratios of $q = 0.5, 0.9, 0.95$ in 7B respectively: $(A = 0.53, k = 0.32)$, $(A = 1.65, k = 0.62)$, $(A = 3.57, k = 0.61)$, and in 14B respectively: $(A = 0.63, k = 0.29)$, $(A = 3.3, k = 0.37)$, and $(A = 1, k = 0.63)$. We hope that this scaling law guides the future study of fine-grained quantization for LLMs.

## 5 DISCUSSION

**Implications on AI Inference Hardware and System Designs** A critical finding of this paper is the observed correlation between model size and quantization ratio, indicating that larger models can accommodate exponentially more low-precision components without performance degradation. This validates the recent trend of increasing support for low-precision arithmetic computation in Deep Learning accelerators such as GPUs and TPUs. For example, the tensor cores of the H100 SXM achieve 3958 TFLOPs when operating at FP8 precision, while the compute capacity is approximately halved to 1979 TFLOPs at FP16 precision (Choquette, 2022). The insight from the first *LLM-MPQ* scaling law, showing that larger LLMs require more low-precision computational resources compared to their smaller counterparts, highlights the **need for increased low-precision resources in future hardware devices** for efficient serving of large models.

Additionally, we've shown through the second *LLM-MPQ* scaling law that finer granularity in mixed-precision approaches enables a higher quantization ratio when the model size is fixed. This insight has direct implications in the design of parallelization strategies for multi-device or multi-node environments (Zheng et al., 2022; Li et al., 2023b). Coarse-grained mixed-precision strategies such as layer-wise mappings can generally be handled as a device allocation problem. Meanwhile, finer-grained mixed-precision approaches such as at the vector/column level necessitate more careful

handling, potentially demanding compiler-level partitioning strategies or even dedicated hardware designs to realize the theoretical performance improvements.

**Extension to Further Architectures and Arithmetic Formats** It is natural to consider whether the observed findings in this work extend to larger LLMs, such as the recently released Llama-3.1-405b (Dubey et al., 2024), although the range of available pre-trained checkpoints is limited, due to the significant cost of training larger models. Additionally, the same trends could be explored in different architectures including Mixture-of-Experts (MoE) models such as Mixtral (Jiang et al., 2024) and multimodal models such as Pixtral (Mistral AI). Finally, further arithmetic formats such as ternary (Chen et al., 2024) and additional configurations from the MXINT (Rouhani et al., 2023a) standard offer opportunities for further exploration. One specific challenge is the quantization approach used in this paper is emulated quantization following Zhang et al., where it incurs more computation than natively supported FP16 inference, hence impedes the evaluation on larger models (eg. 400B). A possible future direction would be to test these scaling laws on large models using actual MXINT4 and FP4 quantization upon the availability of compatible hardware.

**LLM Evaluation: Navigating Layers of Complexity** A number of challenges were faced during experimentation regarding the reproducibility of accuracy and perplexity metrics for pre-trained models. For example, the evaluation methodology for Llama-2 was extrapolated from the official repository [3] since the official evaluation code was not released, leading to a gap between the reported performance and our own evaluation. These discrepancies highlight the importance of open and reliable benchmarks for pretrained language models [4].

An additional observation was that despite the breadth of downstream tasks used to evaluate LLMs in the literature, not all are effective in capturing the scaling trend of LLM performance at various quantization methodologies. Some widely reported metrics, such as Wikitext2 and LAMBADA (Paperno et al., 2016), showed negligible sensitivity to the quantization ratio in performance degradation across the models we evaluated, showing that the bias introduced by various downstream tasks needs to be carefully considered when searching for the optimal quantization strategy for deployment. The core results in this work were reported using a subset of the SlimPajama dataset, as this led to a higher sensitivity to quantization ratio compared to instruction-tuning datasets such as Alpaca, as shown in Appendix C. Another important reason for this decision was to ensure that quantization search was performed under a similar data distribution to common LLM pretraining datasets.

**Hypotheses on other Efficient AI Methods** While we focused primarily on mixed-precision quantization in this work, a clear direction for future research involves examining scaling trends for other AI efficiency methods, such as sparsity and low-rank approximations. We hypothesize that the scaling laws for such methods will closely resemble the scaling laws for quantization introduced in this work. More broadly, we hypothesize the existence of **a broader scaling law governing how the ratio of approximate compute to exact compute scales with model sizes**, and the granularity at which approximate compute is applied.

## 6 CONCLUSION

In this paper, we present two scaling laws of the mixed precision quantization of LLMs verified by extensive experiments, *i.e.*, *LLM-MPQ* scaling law 1) the quantization ratio for a fixed loss target exponentially scales with the model size. 2) The max quantization ratio achievable for a given loss target increases with finer quantization granularity. These two laws offer a guidance to further studies on LLM quantization, and indicate a potential scaling trend for designing low-precision LLM inference accelerators.

---

[3]Official repository for the LLaMA model: github.com/meta-llama

[4]Our current benchmark relies on `lm-eval-harness`, which does not include implementations for all relevant benchmarks.

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

## A    SLIMPAJAMA PERPLEXITY FOR DIFFERENT MODEL FAMILIES

In this section, we show the complete set of results for SlimPajama perplexity and percentage change $\Delta ppl$ compared to the non-quantized baseline for QWen1.5, Llama2 and Gemma2 models across a range of quantization ratio $Q_r$. The ratios are selected in log scale intervals to give an overview of the quantization ratio space and focus on the maximal ratio achievable across models. Note that the resolution for layer-wise mixed quantization is coarser than that for matmul-wise, due to the reduced granularity. Specifically, we evaluate up to a ratio of 0.99 for matmul-wise quantization.

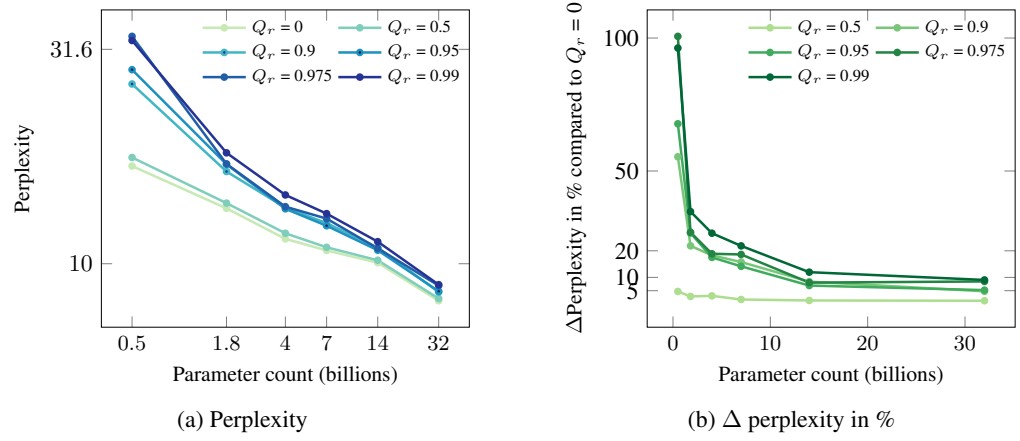

(a) Perplexity

(b) $\Delta$ perplexity in %

Figure 9: `qwen` on `pajama` in matmul-wise.

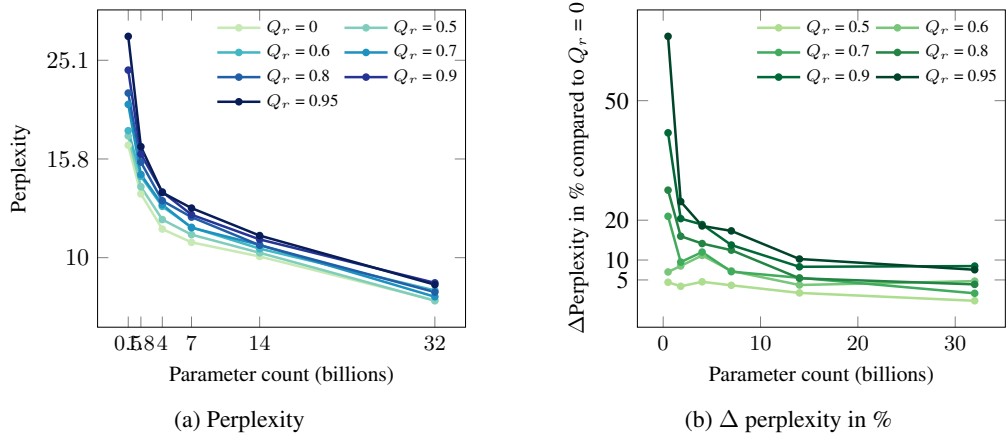

(a) Perplexity

(b) $\Delta$ perplexity in %

Figure 10: `qwen` on `pajama` in layer-wise.

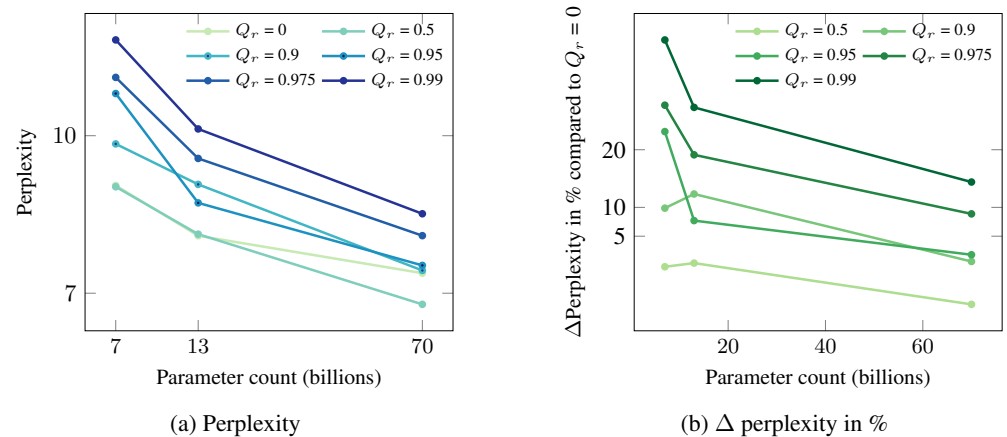

(a) Perplexity

(b) $\Delta$ perplexity in %

Figure 11: `llama2` on `pajama` in matmul-wise.

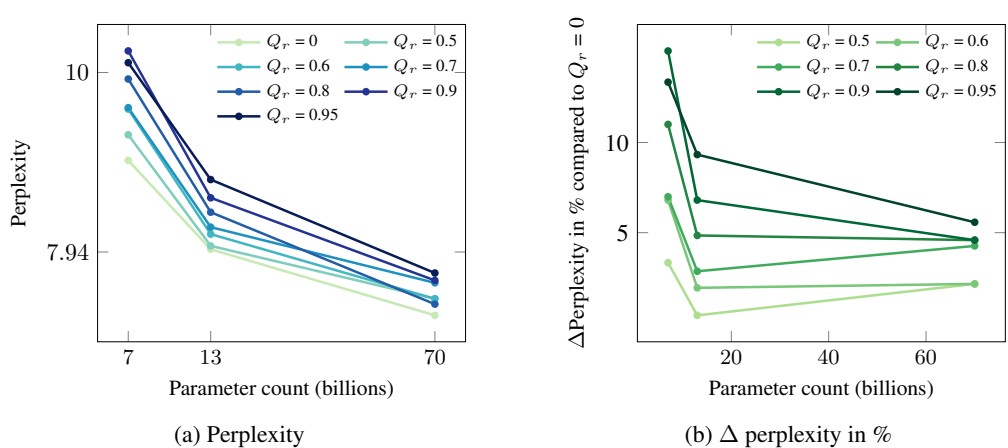

(a) Perplexity

(b) $\Delta$ perplexity in %

Figure 12: `llama2` on `pajama` in layer-wise.

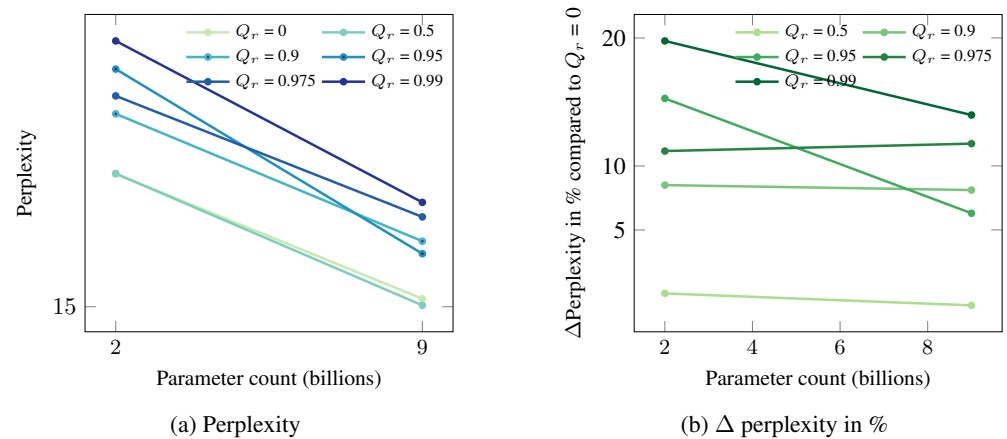

(a) Perplexity

(b) $\Delta$ perplexity in %

Figure 13: `gemma2` on `pajama` in matmul-wise.

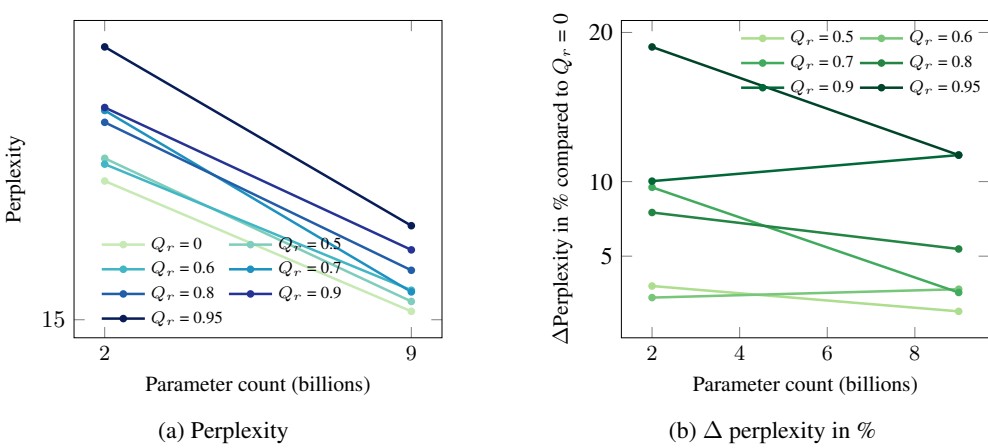

(a) Perplexity

(b) $\Delta$ perplexity in %

Figure 14: `gemma2` on `pajama` in layer-wise.

## B   DOWNSTREAM TASK METRIC FOR QWEN1.5

In this section, we show the complete set of evaluation (more $Q_r$ ratios) for the MMLU downstream task. We report the MMLU evaluation accuracy for Qwen 1.5 models, as well as $\Delta$accuracy.

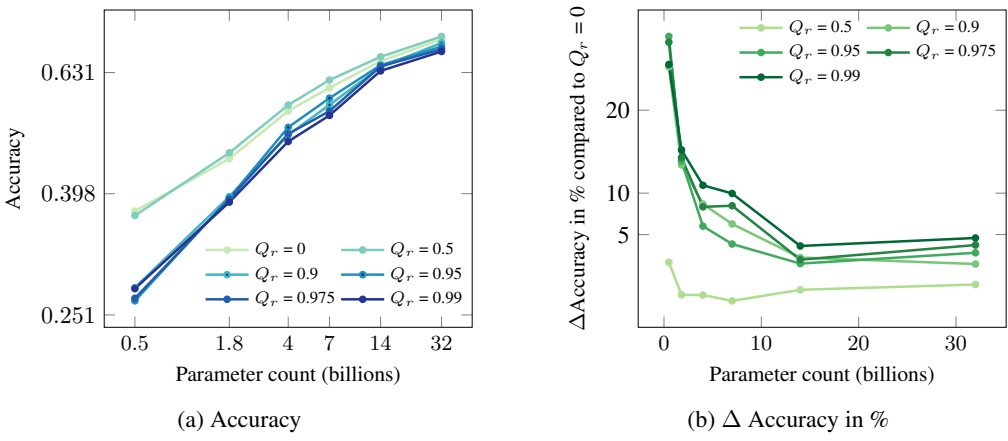

(a) Accuracy

(b) $\Delta$ Accuracy in %

Figure 15: `qwen` on `mmlu` in matmul-wise.

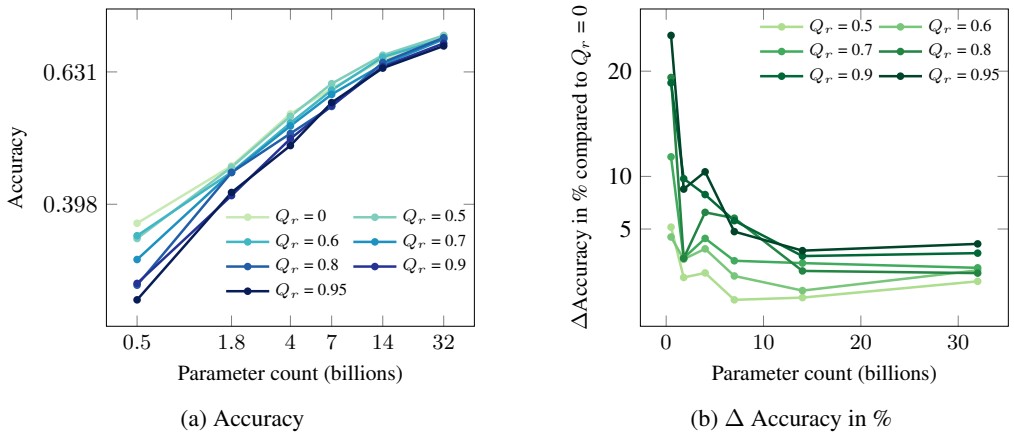

(a) Accuracy

(b) $\Delta$ Accuracy in %

Figure 16: `qwen` on `mmlu` in layer-wise.

# C  COMPARING ALPACA AND SLIMPAJAMA SEARCH PERPLEXITY

In this section, we show that the observations made in this paper are general and extend to other pre-training datasets such as Alpaca. Figure 17 shows similar results for Alpaca compared to SlimPa-jama in Figure 18.

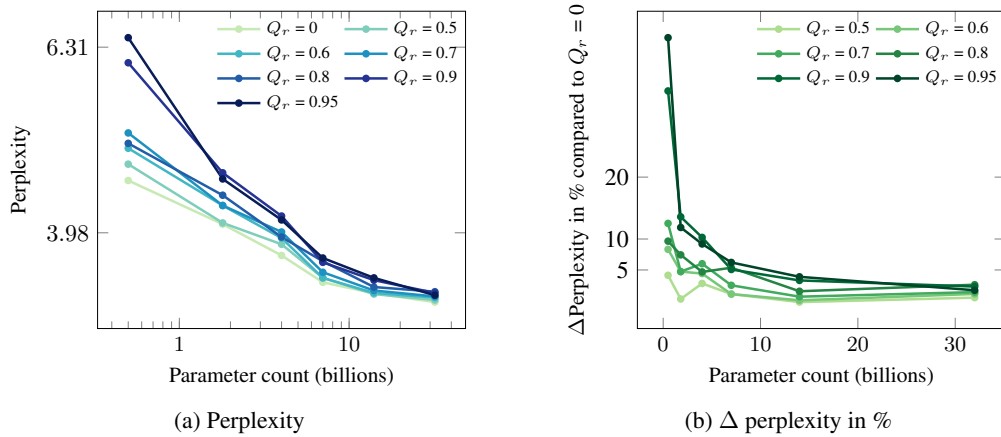

(a) Perplexity

(b) $\Delta$ perplexity in %

Figure 17: `qwen` on `alpaca` in layer-wise.

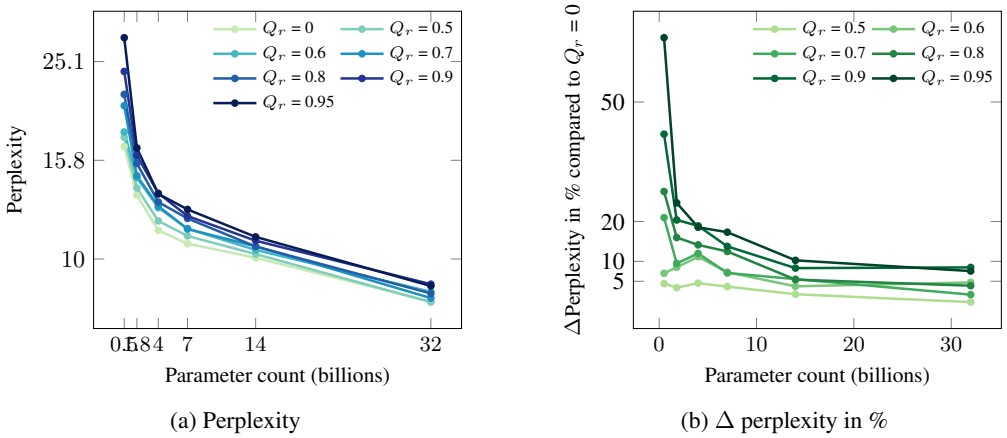

(a) Perplexity

(b) $\Delta$ perplexity in %

Figure 18: `qwen` on `pajama` in layer-wise.

## D   AN ABLATION STUDY ON NUMBER OF TRIALS

To illustrate the selection for our trial number for the random search, we demonstrate the result for setting the trial numbers to 10, 20, 50, 100, and 200 in a random search on QWen1.5 7B model with $Q_r = 0.9$. As shown in Figure 19, our selection of 50 search trails reaches similar results with longer trails. Hence, it is selected as the trial number for our search.

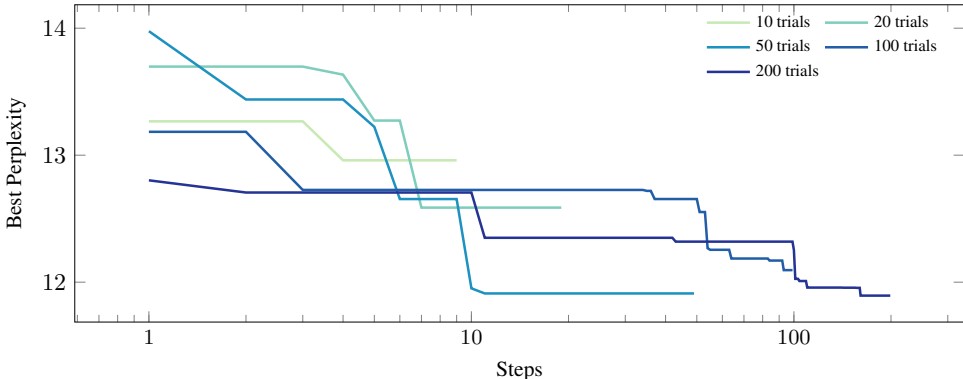

Figure 19: Best perplexity reached with given search trails.

## E   AN ABLATION STUDY ON NUMBER OF SUB-SAMPLES

To illustrate the selection for our sub-sample sizes for the random search, we show the effect on perplexity values over different numbers of sub-samples for the QWen-1.5-7B model on SlimPajama. In Figure 20, the curve saturates at 1024 and is selected as the number of samples for our search.

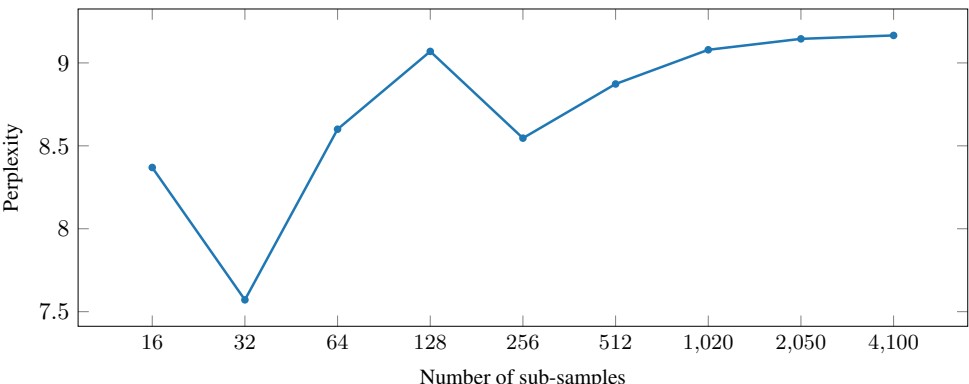

Figure 20: Best perplexity reached under given sub-sample size.

# F LAMBADA RUNS WITH OPT FAMILY

Here we show that not all downstream tasks effectively reflect the performance of quantized LLMs, especially for older models, such as the OPT family. Figure 21 shows our results of OPT family on the LAMBADA (Paperno et al., 2016). The loss of accuracy is negligible when scaling to larger model sizes.

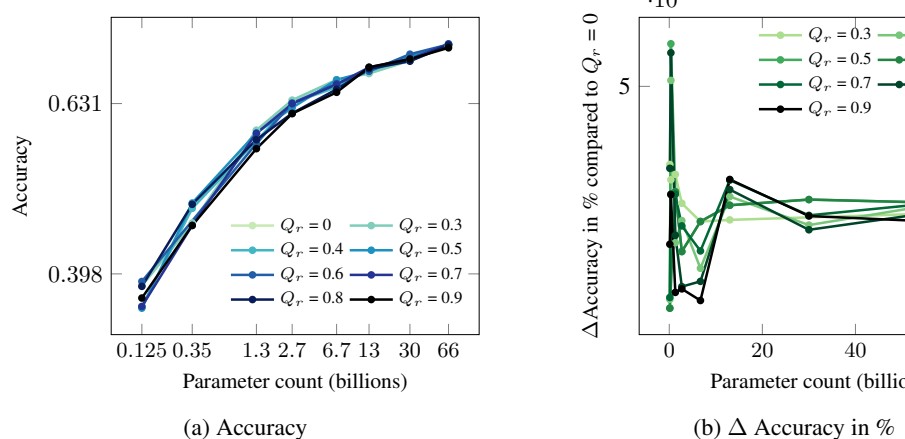

(a) Accuracy

(b) Δ Accuracy in %

Figure 21: `qwen` on `pajama` in layer-wise.

