# OpenReview forum: "Scaling Laws For Mixed Quantization In Large Language Models"
_ICLR.cc/2025/Conference — ICLR 2025 Conference Withdrawn Submission_

### Official Review · Reviewer_mYSh · 2024-11-02

**Soundness:** 2
**Presentation:** 3
**Contribution:** 2
**Rating:** 3
**Confidence:** 5

**Summary:**

This paper examines mixed-precision quantization (Post-Training Quantization, PTQ) for large language models (LLMs) and proposes scaling laws that describe the relationship between model size, quantization ratio, and performance. By mixing low and high precision parameters, the study focuses on reducing the memory and computational costs of LLMs, systematically analyzing efficient quantization strategies for large models.

**Strengths:**

The paper includes a thorough literature review, making it easier for readers to follow up on relevant studies.

It experimentally confirms existing knowledge, providing empirical support for well-known intuitions.

**Weaknesses:**

The finding that larger models tolerate higher compression ratios could be intuitively or numerically inferred from previous studies, suggesting limited novelty.

The paper quantifies trends through curve fitting or regression on specific models, which may not yield fundamentally new insights.

**Questions:**

1.	To what extent can the proposed scaling laws be generalized across different models?
2.	The result of curves may vary significantly depending on the specific quantization technique used.
3.	Could the authors provide more concrete details on how the proposed laws might directly influence hardware design? How can we utilize the law in real situation?

---

### Official Review · Reviewer_ErvP · 2024-11-02

**Soundness:** 2
**Presentation:** 1
**Contribution:** 2
**Rating:** 3
**Confidence:** 3

**Summary:**

the authors try to find the limits of preserving LLM quality after pushing LLMs into lower bits. They define quantization ratio to gain insight into the scaling laws through their experiments. The authors present two scaling laws, one for the model and another for quantization granularity. They ran extensive experiments and traced the change in perplexity and quantization ratio along with increasing parameter size.
I believe the paper lacks proper structure. They simply follow the existing scaling laws for standard FP32. However, going to lower bits bring its own challenges to learning from the scaling laws. This paper brings a limited conclusion.

**Strengths:**

This is one of the rare papers that tries to bring insights about scaling laws in terms of bit width; however, still, they study the model parameters to scale common models. The paper offers a guide for further studies in Conclusion section through their discovered scaling laws with limited impact.
I waited to see a study on scaling laws in low-precision arithmetic, and I was excited to see this paper that tries to push the boundaries. The paper attacks an important problem and provides preliminary results toward this goal but lacks proper structure and the conclusion after this study remained limited.

**Weaknesses:**

An extensive scaling law for quantized LLMs require a structured study that this paper lacks. I recommend structure the problem first
 before studying the scaling laws in lower bits and tell us where we are increasing our knowledge through these experiments.

1- Study i) quantized weights, ii) quantized weight and activation separately. They lead to different scaling laws.
2- Study post-training quantization, vanilla zero shot such as Round To Nearest(RTN), vanilla oneshot such as GPTQ,  and compare it with quantize-aware training separately. They lead to different scaling laws.
3- Study number formats low bit floating points with different numbers of bits, different mantissa, different exponent bits, and, if possible, different accumulator sizes. Study on fixed point and dynamic fixed point, and group size for quantization. Each of these settings provides a different scaling curve.

Maybe after having this big picture we can conclude how these scaling laws are related, and how they differ from the common FP32, FP16, BF16 scaling laws in language models. The optimal frontier in all these studies perhaps provide us insights about what are the limits of lower-bit scaling compared with FP32, FP16, BF16.

Study on scaling laws idealy must have three different objectives: i) pre-training  ii) training (fine-tuning) iii) inference. The scaling laws are often understood in the pre-training phase. I hope that more recent papers shed some light on the other two parts and help us to fill the gap of low-bit parameter setting with standard FP32, FP16, and BF16 number format.

Overall, I think the paper formulate the problem properly first. I had the feeling that this paper is valuable only because of the experiments and conclusion from these experiments are not much impactful.

**Questions:**

- Lower-bit quantization on weights is often performed to save data transfer from the register bits, and lower bit activation is often studied to speed up the arithmetic at inference  (and sometimes for gradient computation). Can you provide scaling laws in terms of thebit size  and replace as a replacement of the model parameter size? If yes, what would be the best combination of weights representation? how many bits, how many exponents, how many mantissa?

- This study looks interesting, especially the idea of designing a new number format by learning from the existing number formats FP4. How can we bring the accumulator into the picture (which is often unknown and hardware-dependent)?

---

### Official Review · Reviewer_rQbe · 2024-11-04

**Soundness:** 2
**Presentation:** 1
**Contribution:** 2
**Rating:** 3
**Confidence:** 4

**Summary:**

This paper focuses on studying the relationship between the quantization ratio and model size in mixed-precision weight-only quantization. Specifically, the authors use a combination of 4-bit and 16-bit quantization at the transformer block and linear layer granularities and conduct experiments across various models and datasets. With extensive experiments, the authors summarize two phenomena: (1) The larger the models, the better they can preserve performance with an increased quantization ratio; (2) The finer the granularity of mixed-precision quantization, the more the model can increase the quantization ratio.

**Strengths:**

1. This paper not only addresses low-bit integer quantization but also explores low-bit floating-point quantization, which may facilitate the deployment of LLMs on the latest hardware.
2. The experiment part is solid, as the author has conducted analyses across a variety of models and datasets.

**Weaknesses:**

1. My main concern is the novelty of this paper is quite limit.

    a. This paper focuses solely on mixed-precision weight-only quantization without addressing activation and KV cache quantization. Therefore, I suggest using "Mixed-Precision Weight-Only Quantization" instead of "Mixed Quantization" in the title.

    b. The two proposed scaling laws for mixed-precision weight-only quantization appear rather straightforward and may not be new to readers. Here are the reasons:

      - (1) For the first scaling law: Larger models retain performance better with reduced bit-width, which has already been extensively studied [1,2]. It is well-known that larger models can tolerate lower bit-widths, making it a fairly straightforward conclusion to apply higher compression ratios to larger LLMs.

      - (2) For the second scaling law: Finer granularity in mixed-precision quantization allows for a higher quantization ratio. This is also a predictable result, as fine-grained quantization can improve accuracy but introduces greater hardware efficiency overhead.

2. The writing in this paper lacks clarity and can be confusing.

    a. The terminology in this paper is inconsistent or inaccurate. Here are some example:

      - Inconsistency: In line 45, the authors define "transformer block," but this term actually refers to "transformer layer" in the abstract.

      - Inconsistency: In line 20, the authors refer to only two granularities, layer-wise and matmul-wise. However, in line 213, they mention per-vector, per-tensor, and per-layer. It appears that "layer-wise and matmul-wise" are used to describe bit-width allocation granularity, while "per-vector, per-tensor, and per-layer" refer to scaling factor sharing granularity. Consistent and clear use of these terms is recommended.

      - Inaccuracy: In line 20, the authors mention "matmul-wise." However, LLMs involve multiple types of matrix multiplications. For example, in the linear layer, **activation and weight** matrices perform general matrix multiplication (GEMM). In the attention mechanism, the query and key **activation** matrices also use GEMM, while in GLU, gate_proj and up_proj outputs undergo element-wise matrix-matrix multiplication. It seems the authors may actually be referring to the first type, linear layer.

    b. In the background section, the connection between the listed topics and this paper is unclear

      - Weight and Activation Outliers in LLMs: The authors do not provide outlier analysis in the method part or the experiment part.

      - Mixed-Precision Training: This paper primarily focuses on post-training quantization, so it’s unclear why mixed-precision quantization-aware training is discussed in detail. It’s recommended to clarify the relevance here.

      - Scaling Laws of Large Language Model Training:  Again, it doesn’t seem necessary to delve into this topic in such detail. This topic is unrelated to both quantization and post-training.

      - In contrast, I think it is necessary to discuss the differences between this paper and existing evaluation work about quantization [1,2,3].

3. The experimental setup needs to be more comprehensive.

    a. Why are only 4-bit and 16-bit applied to weights? A wider range of bit-width combinations is needed. For example, consider using a combination of 16-bit, 8-bit, 4-bit, and 2-bit.

    b. It is recommended to include some challenging tasks, such as MT-Bench (multi-turn conversation).

  [1] Yao, Zhewei, et al. "Zeroquant-v2: Exploring post-training quantization in llms from comprehensive study to low rank compensation." arXiv preprint arXiv:2303.08302 (2023).
  [2] Li, Shiyao, et al. "Evaluating quantized large language models." arXiv preprint arXiv:2402.18158 (2024).
  [3] Liu, Peiyu, et al. "Do emergent abilities exist in quantized large language models: An empirical study." arXiv preprint arXiv:2307.08072 (2023).

**Questions:**

See weaknesses

---

### Official Review · Reviewer_F8kv · 2024-11-04

**Soundness:** 2
**Presentation:** 2
**Contribution:** 2
**Rating:** 3
**Confidence:** 3

**Summary:**

The authors conducted extensive experiments to study the scaling law of the model's perplexity with respect to maximum quantization ratio(number of parameters in low precision over number of total parameters) and quantization granularity.

**Strengths:**

1. Extensive experiments conducted
2. The authors were able to parametrize the exponential relationship between perlexity change and quantization ratio, and the power relationship between the perplexity change and quantization granularity.

**Weaknesses:**

1. Confusion for granularity. After reading the paper, I feel confused by what granularity is refering to. In the abstract, section 2.1 and discussion, the granularity is refering to layer-wise or matmul-wise. In section 4.3, the scaling law derived from quantization granularities is the block size. In general, I suggest the authors to be more consistent in their writing.
2. While finer granularity enables higher quantization ratio, it also requires more memory since more quantization scales are needed due to smaller block sizes. How do the authors address this?
3. Inconsistency for scaling law formulation. In section 2.2, the scaling laws are formulated where the "y" term is the maximum acheivable mixed precision quantization ratio, while in section 4, the "y" term is the change in perplexity.

**Questions:**

1. Can the authors provide more details on how the random search is conducted?
2. For per-layer quantization ratio, are all the tensors in the same layer quantized using the same quantization scale? Or the quantization is performed per tensor while the whole layer is quantized using the same format (high or low precision)?

**Details Of Ethics Concerns:**

None.

---

### Note · Authors · 2024-11-13

I have read and agree with the venue's withdrawal policy on behalf of myself and my co-authors.